# Identification of Heparin-Induced Thrombocytopenia in Surgical Critically Ill Patients by Using the HIT Expert Probability Score: An Observational Pilot Study

**DOI:** 10.3390/jcm11061515

**Published:** 2022-03-10

**Authors:** Guillaume Besch, Dejan Ilic, Marc Ginet, Clément d’Audigier, Philippe Nguyen, David Ferreira, Emmanuel Samain, Guillaume Mourey, Sebastien Pili-Floury

**Affiliations:** 1Department of Anesthesiology and Intensive Care Medicine, University Hospital of Besancon, F-25000 Besancon, France; dilic@chu-besancon.fr (D.I.); mginet@chu-besancon.fr (M.G.); dferreira@chu-besancon.fr (D.F.); e1samain@chu-besancon.fr (E.S.); spilifloury@orange.fr (S.P.-F.); 2EA3920, University of Franche-Comte, 3 bvd Alexander Fleming, F-25000 Besancon, France; 3Etablissement Français du Sang Bourgogne Franche-Comté, Hemostasis Department, Laboratoire de Biologie Médicale et de Greffe, F-25000 Besançon, France; clement.daudigier@efs.sante.fr (C.d.); guillaume.mourey@efs.sante.fr (G.M.); 4Department of Haematology, CHU Robert Debré, F-51000 Reims, France; pnguyen@chu-reims.fr; 5EA3801, IFR-53, Reims Champagne-Ardenne University, F-51000 Reims, France; 6EA481, Integrative and Clinical Neuroscience Laboratory, University Hospital of Besancon, F-25000 Besancon, France; 7Department of Clinical Hemostasis, University Hospital of Besancon, F-25000 Besancon, France

**Keywords:** heparin, thrombocytopenia, critical care, heparin/adverse effects, thrombosis

## Abstract

Background: Heparin-induced thrombocytopenia (HIT) remains a challenging diagnosis especially in surgical intensive care unit (SICU) patients. The aim of the study was to evaluate for the first time the diagnostic accuracy of the HIT Expert Probability (HEP) score in the early identification of HIT in SICU patients. Methods: The HEP and 4Ts scores were calculated in all patients with suspected HIT during their stay in our SICU. The diagnosis of HIT was finally confirmed (HIT+ group) or excluded (HIT− group) by an independent committee blinded to the HEP and 4Ts score values. The primary outcome was the sensitivity and specificity of a HEP score ≥ 5 for the diagnosis of HIT. The secondary outcome was the area under the ROC curve (AUC) of the HEP and 4Ts scores in the diagnosis of HIT. Results: Respectively 6 and 113 patients were included in the HIT+ and HIT− groups. A HEP score value ≥ 5 had a sensitivity (95% confidence interval (95% CI)) of 1.00 (0.55–1.00), and a specificity (95% CI) of 0.92 (0.86–0.96). The AUC (95% CI) was significantly higher for the HEP score versus for the 4Ts score (0.967 (0.922–1.000) versus 0.707 (0.449–0.965); *p* = 0.035). Conclusions: A HEP score value < 5 could be helpful to rule out HIT in SICU patients.

## 1. Introduction

Patients admitted to the surgical intensive care unit (SICU) have a high risk of thromboembolic events, justifying the prescription of heparin. Heparin-induced thrombocytopenia (HIT) is a well-known complication of heparin therapy caused by platelet-activating antibodies binding to PF4/heparin complexes. HIT is quite rare in SICU patients (overall incidence < 2%) [1,2] but early recognition of HIT is crucial because of the high morbidity and mortality related to arterial and venous thrombosis [3]. Rapid and early diagnosis of HIT is still difficult. The initial presentation of HIT tends to be limited to the development of thrombocytopenia in a patient under heparin treatment. Laboratory tests prescribed to confirm or rule out HIT suffer from limitations. Immunoassays yield rapid results, with high sensitivity, but with insufficient specificity. Functional assays are unavailable in many hospitals and their sensitivity was recently called into question by expert groups [4,5]. The diagnosis of HIT remains particularly challenging in SICU patients since the prevalence of thrombocytopenia could be as high as 50% [6,7], leading to frequent overdiagnosis and overtreatment [2,6]. The unnecessary discontinuation of anticoagulation exposes the patient to the risk of thrombosis [7], while non-heparin anticoagulation prescribed in suspected HIT is expensive and increases the risk of bleeding [8,9,10,11]. Finally, confirming the diagnosis of HIT with certainty remains difficult, requiring close collaboration between attending and hemostasis physicians.

Lo et al. proposed combining clinical and biological data in an easy-to-use score comprising four items, to assess the pretest probability of HIT, namely, the 4Ts score [12]. A 4Ts score value ≤ 3 has a high negative predictive value, which makes it possible to rule out the diagnosis of HIT [12,13]. However, the performance of the 4Ts score has been challenged in critically ill patients because of unacceptably high rates of false-negative and false-positive results reported in some studies [1,2,14,15,16]. Taking into account the limits of the 4Ts score, Cuker et al. developed the HIT Expert Probability (HEP) score [17,18]. The 4Ts and HEP scores are based on items assessing the timing and severity of thrombocytopenia, the onset and timing of thrombosis, the clinical signs of HIT and the potential other causes of thrombocytopenia (see Appendix A, Table A1 and Table A2, respectively). The HEP score differs from the 4Ts score in that it weights each item with either positive or negative points [17,18]. In particular, the HEP score allows for the attribution of negative points, which may reach −10 when other causes of thrombocytopenia are present, a common situation in SICU patients. To date, the HEP score has never been studied in SICU patients. We hypothesized that the HEP score would be more accurate than the 4Ts score to assess the pretest probability of HIT in SICU patients, by assigning negative points to alternative causes of thrombocytopenia.

The aim of this study was therefore to evaluate the diagnostic accuracy of the HEP score for the early identification of HIT in surgical critically ill adult patients. The secondary objective was to compare the diagnostic performance of the 4Ts and HEP scores.

## 2. Materials and Methods

### 2.1. Study Design

A prospective, single center, observational pilot study was conducted in the SICU of the University Hospital of Besançon, France. The study protocol was approved by the local Institutional Review Board of the University Hospital of Besancon, France (Chairperson: Mr. J. Deglise), on 12 February 2012. The study was conducted in accordance with the French bioethics law (Art. L. 1121-1 of the law no. 2004-806, 9 August 2004).

### 2.2. Participants

All consecutive patients admitted to the SICU of the University Hospital of Besançon, France, between October 2012 and October 2014 with suspected HIT were eligible. The diagnosis of HIT was suspected by the attending physician when at least one of the following criteria was present [19]: (1) thrombocytopenia < 100 G/L and/or a platelet count decrease > 40%; (2) clinical and/or biological abnormalities consistent with HIT between 5 and 8 days after initiation of unfractionated or low molecular weight heparin treatment (primo-exposure), or within 5 days in case of heparin exposure during the previous 3 months; (3) arterial and/or venous thrombosis, atypical hemorrhage, or skin reaction to subcutaneous heparin injection; (4) extension or recurrence of arterial and/or venous thrombosis despite well-conducted heparin anticoagulation. Exclusion criteria were: age < 18 years, pregnancy and/or breast-feeding, legal inability or refusal to provide informed consent, or anticoagulation with fondaparinux. Patients received oral and written information before inclusion in the study. If the patient was unconscious or unable to consent, the oral and written information was delivered to the next of kin, and patient consent was obtained when possible. Patients were included on the day when HIT was first suspected.

### 2.3. Data Collected and Endpoint Measurements

Baseline characteristics, the reason for admission to the SICU, the type of heparin (unfractionated versus low molecular weight heparin), the reason for heparin therapy, the time from initiation of heparin therapy to suspicion of HIT, and all daily platelet count values from admission to discharge from the SICU were extracted from medical files.

The 4Ts and the HIT Expert Probability (HEP) scores (see Appendix A, Table A1 and Table A2, respectively) were calculated on the day of inclusion, and the following blood analyses were performed: (1) rapid particle gel immunoassay ID-PaGIA (Diamed GmbH, Cressier-sur-Morat, Switzerland); (2) detection of IgG HIT antibodies using enzyme-linked immunosorbent assay (ELISA) (Hyphen Biomed, Neuville-sur-Oise, France); (3) heparin-induced Platelet Aggregation Test (PAT) (SODEREL Aggregometer, Nancy, France); and (4) platelet serotonin-release assay (SRA). All blood analyses were conducted in the Hemostasis Department of the French Blood Transfusion Centre of Bourgogne Franche-Comte, Besançon, France, except for SRA, which was performed in the Hemostasis Department of the University Hospital of Reims, France. From blood sampled in each patient, plasma aliquots were stored at −20 °C in the Hemostasis Department of the French Blood Transfusion Centre (Etablissement Français du Sang, EFS) of Bourgogne Franche-Comte, Besançon, France, to perform ELISA, PAT, and SRA tests at the end of the study inclusion period. Symptomatic and asymptomatic deep vein thrombosis was investigated with Doppler ultrasound imaging of the upper and lower limb veins in each patient included, except when the HIT was suspected on the basis of a new thromboembolic event under heparin therapy (deep vein thrombosis, pulmonary embolism).

Discontinuation of heparin therapy and initiation of non-heparin anticoagulation were decided by the attending physician. Briefly, the decision was based on the 4Ts score value, Doppler ultrasound imaging, and ID-PaGIA, and was finally adjusted after receiving the results of the ELISA and PAT. The attending physician was blinded to the HEP score value. The type of non-heparin anticoagulation (none, argatroban, lepirudin, and danaparoid sodium), the complications of non-heparin anticoagulation, the length of stay in the SICU, the duration of mechanical ventilation, and the vital status at discharge from the SICU (dead or alive) were also recorded.

Finally, the diagnosis of HIT was made by an independent committee, comprised of 3 intensivists and the head of the Hemostasis Department of our institution. The decision was based on the course of the clinical situation, the kinetics of the platelet count (with and/or without discontinuation of heparin therapy), and the results of the ELISA, PAT, and SRA. The committee was blinded to the 4Ts and HEP scores and the results of the ID-PaGIA.

The primary objective was to assess the diagnostic accuracy of the HEP score for the early identification of HIT in surgical critically ill adult patients. The primary endpoint was the Sensitivity (Se), specificity (Sp), positive (PPV), and negative (NPV) predictive values of the HEP score for the diagnosis of HIT. The secondary objectives were to assess the incidence of HIT in our surgical intensive care unit, the diagnostic accuracy of the 4Ts score and the ID-PaGIA in this population to compare the diagnostic performance of the 4Ts and HEP scores.

### 2.4. Test Methods

#### 2.4.1. The 4Ts Score

The 4Ts score was calculated on the day of inclusion as follows [10]: 2 points were attributed for thrombocytopenia with a platelet count decrease > 50% and platelet nadir ≥ 20 G/L and 1 point for thrombocytopenia with a platelet count decrease 30% to 50% and platelet nadir 10 to 19 G/L; 2 points were attributed if the platelet decrease clearly happened between days 5 and 10 of heparin therapy or if platelets decreased ≤ 1 day, and 1 point if the platelet decrease occurred after day 10; 2 points were attributed in case of confirmed thrombosis and 1 point in case of recurrent, progressive or suspected thrombosis; 2 points were added if another cause of thrombocytopenia was unlikely and 1 point if possible other causes of thrombocytopenia were present. The sum of the points yields the 4Ts score, resulting in a pretest probability of HIT assessed as high (6–8 points), intermediate (4–5 points), and low (≤3 points) (see Appendix A, Table A1).

#### 2.4.2. The ID-PaGIA (Particle Gel Immuno Assay) Heparin/PF4 Antibody Test

The ID-PaGIA is a rapid particle gel immunoassay that allows the detection of IgG, A, and M specific to heparin/PF4 complexes. The ID-PaGIA was performed according to the manufacturer’s instructions (Diamed/Biorad^®^ GmbH, Cressier, Switzerland). Briefly, 10 µL of plasma was mixed with 50 µL of polymer particles coated with heparin/PF4 complexes in the reaction chamber of the test ID-card. The ID-card was incubated at room temperature for 5 min and then centrifuged for 10 min in the appropriate ID-centrifuge (Diamed SA). The ID-PaGIA was performed on the day of inclusion, and the result was available a few hours after blood sampling, as follows: positive (in favor of HIT), if the particles aggregated at the top of the gel chamber; negative if the particles sank to the bottom of the gel chamber; or intermediate if the particles neither clearly agglutinated at the top nor fully settled at the bottom of the gel chamber.

#### 2.4.3. The Zymutest HIA IgG (ELISA)

The ELISA test (Hyphen Biomed, Neuville-sur-Oise, France) was performed once a week in the Hemostasis laboratory of our institution on the blood sample drawn on the day of inclusion. The result of the ELISA was available for the attending physician once a week, as follows: positive (in favor of HIT); negative; or intermediate. The results of all the tests performed were available to the independent committee to make the diagnosis of HIT.

#### 2.4.4. The Heparin-Induced Platelet Aggregation Test (PAT)

The PAT is a HIT platelet functional assay. Platelet Rich Plasma (PRP) from 4 healthy blood donors was added to plasma sampled from patients with suspected HIT in the presence of heparin 1 IU/mL. The appropriate response of the platelets of the 4 healthy blood donors was previously tested by adding the plasma from a patient with confirmed HIT. Platelet response to the patient plasma was quantified by using a Light Transmission Aggregometer (SD-Innovation, Frouard, France) and expressed as the maximal percentage of aggregation. The PAT was positive if: (1) genuine aggregation was observed, similar to that observed with the HIT positive plasma; and (2) inhibition of this platelet aggregation while adding heparin 100 IU/mL. All PAT were performed at the end of the inclusion period in the Hemostasis laboratory of our institution on the blood sample drawn on the day of inclusion.

#### 2.4.5. The [^14^C]-Serotonin Release Assay (SRA)

The SRA is a complex platelet functional assay. The SRA is only performed in reference laboratories that are licensed to handle radioactive substances. All SRA were performed at the end of the inclusion period in the Hemostasis laboratory of the University Hospital of Reims, France, with the dedicated blood sample drawn on the day of inclusion. The SRA results were reported as positive if ≥20% release of [^14^C]-serotonin with low dose of heparin and <20% release of [^14^C]-serotonin with high dose of heparin were observed. Negative SRA were reported if both low-dose and high-dose heparin concentrations demonstrated less than 20% serotonin release.

#### 2.4.6. The Index Test: The HIT Expert Probability (HEP) Scores

The HIT Expert Probability (HEP) score was calculated according to Cuker et al. [15] on the day of inclusion by an independent investigator, blinded to the 4Ts score and the results of the following blood analyses: ID-PaGIA, ELISA, PAT, and SRA. The HEP score ranged from –16 to + 19, by summing positive or negative points assigned to the following items: (1) the magnitude of the fall in platelet count; (2) the timing of fall in platelet count; (3) the nadir platelet count; (4) suspected or confirmed thrombosis; (5) skin necrosis at subcutaneous heparin injection sites; (6) acute systemic reaction after intravenous heparin injection; (7) the presence of bleeding, petechiae or extensive bruising; and (8) the existence of other causes of thrombocytopenia (see Appendix A, Table A2).

#### 2.4.7. The Standard Reference Diagnosis of HIT

Since no clinical or biological test is currently able to ascertain the diagnosis of HIT with 100% sensitivity, the final diagnosis of HIT was based on the decision made by an independent committee. The independent committee was composed of 3 intensivists and the head of the department of the Hemostasis Department of the French Blood Transfusion Centre (Etablissement Français du Sang, EFS) of Bourgogne Franche-Comte, Besançon, France. All medical files were reviewed by the independent committee at the end of the inclusion period. The course of the patient’s clinical situation and the kinetics of the platelet count with and/or without discontinuation of heparin therapy, and the results of the ELISA, PAT, and SRA, were presented patient by patient during a dedicated meeting of the independent committee. The committee was blinded to the 4Ts and HEP scores and to the results of the ID-PaGIA. Each member of the committee provided his/her own decision patient by patient and was blinded to the decision made by the others. The committee finally decided with an absolute majority of its members. The rules for the final decision making by the independent committee were laid down in the study protocol before the adjudication meeting.

### 2.5. Analysis

Patients were classified into 2 groups: the HIT+ group, if the final decision of the independent committee was “confirmed HIT”, and the HIT− group when the independent committee ruled out this diagnosis.

Data are presented as median (interquartile range) and number of patients (percentage) for continuous and categorical variables respectively. Intergroup comparisons were conducted by using the Mann–Whitney *U* test and the Fisher’s exact test for continuous and categorical variables respectively.

Sensitivity, specificity, positive and negative predictive values, and positive and negative likelihood ratios of the 4Ts score, ID-PaGIA, ELISA, and SRA were calculated. As the ID-PaGIA could be “indeterminate”, 3 sensitivity analyses were performed: (1) “indeterminate” ID-PaGIA excluded from the analysis, (2) “indeterminate” ID-PaGIA considered as positive, and (3) “indeterminate” ID-PaGIA considered as negative. The area under the ROC (Receiver Operating Characteristic) curves of the 4Ts and HEP scores were compared using the method of DeLong et al. [20]. HEP score was dichotomized on the basis of the value that maximized the sensitivity. The sensitivity, specificity, positive and negative predictive values, and positive and negative likelihood ratios of the dichotomized HEP score were then calculated.

All statistical analyses were performed with SAS software, version 9.4 (SAS Institute Inc., Cary, NC, USA), and the significance level was fixed at 0.05.

## 3. Results

### 3.1. Participants

Among the 1334 patients admitted to the ICU during the study period, 119 had suspected HIT, and 6 (5%) had confirmed HIT (HIT+ group) (Figure 1).

All final decisions made by the independent committee were unanimous. HIT was suspected from: thrombocytopenia in 73 (61%) patients; thrombosis in 29 (24%) patients; thrombocytopenia and thrombosis in 16 (13%) patients; heparin resistance in 1 (1%) patient. The overall incidence of HIT during the study period was 0.43%. Baseline characteristics and outcomes of patients in the 2 groups, are presented in Table 1.

All patients in the HIT+ group (argatroban: 1 (17%) patient; lepirudin 1 (17%) patient; and danaparoid sodium: 4 (67%) patients) versus 16 (14%) patients in the HIT− group (argatroban: 0 patient; lepirudin 6 (5%) patients; and danaparoid sodium: 10 (9%) patients) received non-heparin anticoagulation (*p* < 0.0001). Two (33%) versus one (1%) patients suffered from hemorrhage requiring blood transfusion under non-heparin anticoagulation in the HIT+ and HIT− groups, respectively (*p* = 0.006).

### 3.2. Test Results

Test results are presented in Table 2.

The HEP score was significantly higher in the HIT+ group (6 (5–9)) than in the HIT− group (1 ((−1)–3)) (*p* = 0.0002). No patient had negative ID-PaGIA or a negative ELISA in the HIT+ group (Table 2). Four (66%) patients had both negative PAT and SRA.

### 3.3. Diagnostic Accuracy of the HEP Score

The diagnostic accuracy of all tests is reported in Table 3.

The HEP score value that maximized the sensitivity was five. The sensitivity and specificity of a HEP score ≥ 5 were 1.00 (95% CI = 0.55–1.00) and 0.92 (95% CI = 0.86–0.96) respectively (Table 3). No patient with confirmed HIT had a HEP score < 5. Among the nine (8%) patients who had a HEP score ≥ 5 in the HIT− group, the ID-PaGIA and the ELISA tests were negative in seven (6%) patients and positive in the two remaining patients.

The area under the ROC curve (AUC) was significantly higher for the HEP score (AUC (95% confidence interval (95% CI)) = 0.967 (0.922–1.000)) versus the 4Ts score (AUC (95% CI) = 0.707 (0.449–0.965)) (*p* = 0.035) (Figure 2).

## 4. Discussion

The results of this single-center, observational, pilot study suggest that the HEP score could have a higher diagnostic accuracy than the 4Ts score for the early identification of HIT in surgical critically ill adult patients. A HEP score < 5 could make it possible to rule out HIT and avoid the prescription of unnecessary blood tests. In patients with a HEP score ≥ 5, a negative ID-PaGIA or ELISA test could rule out the diagnosis of HIT.

First described by Lo et al. [10], the 4Ts score has come to be widely accepted as a screening tool to avoid futile blood tests and inappropriate heparin discontinuation by rejecting HIT in patients with a low pretest probability of HIT (4Ts score value ≤ 3) [13,21]. Nonetheless, the diagnostic accuracy of the 4Ts score has been challenging in critically ill patients [1,2,14,15]. Intermediate, high, and combined intermediate and high pretest probability 4Ts scores presented quite a low positive predictive value for the diagnosis of HIT [1,2,15]. Although the negative predictive value of low pretest probability remained close to 100% in critically ill patients, the 4Ts score does not make it possible to rule out the diagnosis of HIT without any risk of screen failure of patients with confirmed HIT in this population. This risk, although extremely low, could reasonably be considered as unacceptable, given the poor prognosis of delayed diagnosis for HIT. The negative predictive value of low pretest probability (4Ts score ≤ 3) observed in the present study was consistent with previously published data. Similar values for sensitivity and specificity of combined intermediate and high pretest probability (4Ts score value ≥ 4) and area under the ROC curve have been observed in surgical ICU patients [2,15]. However, some authors reported higher diagnostic accuracy of the 4Ts score in ICU patients, including higher positive predictive value and better sensitivity and specificity of intermediate and high pretest probabilities [1,22]. Several reasons could explain these discrepancies. First, the positive predictive value depends on the prevalence of confirmed HIT [21]. Second, poor inter-observer reliability has been reported with the 4Ts score [12,14]. Third, the characteristics of ICU patients varied from one study to another, with a variable rate of potential coexisting causes of thrombocytopenia.

Indeed, the main limit of the 4Ts score lies in the item “other cause of thrombocytopenia”. The coexistence of other potential causes of thrombocytopenia is frequent, even in patients with confirmed HIT leading to an underestimation of the pretest probability of HIT by calculating the 4Ts score value [2,15,23,24]. The HEP score proposed by Cuker et al. [17] allows for the attribution of negative points, which can reach −10 when other causes of thrombocytopenia are present (see Appendix A, Table A2). In this first study, including 60% surgical patients, Cuker et al. reported a higher area under the ROC curve of the HEP score compared to the 4Ts score [17]. A HEP score < 2 had higher specificity than a 4Ts score < 4 as a screening tool [17]. The sensitivity and specificity of a HEP score ≥ 5 were very close to the values observed in the present study [17]. These results were confirmed in the external validation study by Joseph et al. [18]. In a cohort including 35% surgical patients and 55% ICU patients, HEP and 4Ts scores were found to have a similar area under the ROC curve, and both a HEP score ≥ 2 and HEP score ≥ 5 had lower specificity than previously described [18]. Since the HEP score differs from the 4Ts score by assigning negative points to other causes of thrombocytopenia, these discrepancies could be explained by the difference in the rate of surgical patients. Surgery could indeed account for thrombocytopenia in many patients, and the diagnostic accuracy of the HEP score could be particularly improved in this population

Confirming HIT remains highly difficult in many cases, and thus several biological assays have been developed to help clinicians. Blood immunoassays, such as ID-PaGIA and ELISA, could be falsely positive in ICU patients related to the acquisition of non-pathologic antibodies to PF4-heparin complexes. In the present study, 8% and 6% unconfirmed HIT patients had positive ID-PaGIA and ELISA tests, respectively. Platelet functional tests are proposed as a second stage approach to confirm HIT in patients with positive immunologic assays. The SRA is often considered as the gold standard for the diagnosis of HIT, but has, in fact, several limitations. First, SRAs are expensive, technically difficult, and only performed in selected specialized laboratories. Second, performing SRAs requires a turnaround time of up to several days or weeks and can lead to a delayed diagnosis of HIT. Third, the diagnostic accuracy of HIT has recently been challenged by some experts [25]. In the present study, the diagnosis of HIT was based on the decision of an independent committee, and only two patients with confirmed HIT had a positive functional test (PAT and SRA). The members of the independent committee are HIT experts in our institution and have reviewed all cases of suspected HIT in SICU patients for several years. Their decisions were made at the end of the study on the basis of the clinical course, the kinetics of the platelet count with and/or without discontinuation of heparin therapy, and the results of the ELISA, PAT, and SRA, as usually performed for suspected HIT in SICU patients in our institution. Even if the methods used to confirm HIT differ from previous studies, we report an incidence of HIT that is in line with that data reported by Crowther et al. [1,14]. In the study by Cuker et al., the diagnosis of HIT was confirmed on the basis of the opinion of three independent experts blinded to the results of the SRA, and the reported sensitivity and specificity of SRA were, respectively, 71% (95% confidence interval (CI): 29–96) and 93% (95% CI: 81–98) for HIT [17]. Part of the false negative SRA could be explained by the potential implication of IgA and IgM antibodies and other cells (such as endothelial cells, monocytes) in HIT, even if their pathogenic role is still a matter of debate [25]. Another reason for the four confirmed HIT with negative SRA could be that HIT may have been suspected very early in the course of thrombocytopenia probably due to the context of the study. The blood level of platelet-activating antibodies could have been too low to activate the platelets during functional tests, and SRA could become positive by repeating the tests. Our study has some limitations. First, since the HEP score was calculated by the intensivist in charge of the patient on the day of the inclusion, interobserver agreement for HEP score was not assessed. Nonetheless, good interobserver reliability has been described by Cuker et al. [17]. Second, Joseph et al. have suggested that the diagnostic accuracy of the HEP score could depend on institutional practice [18] and the results of the present study need to be confirmed in a large multicenter study.

## 5. Conclusions

In our study, the HEP score appeared to have a higher diagnostic accuracy than the 4Ts score in SICU patients. A HEP score < 5 could be helpful to reject HIT in SICU patients. Immunologic assays could be reserved for patients with a HEP score ≥ 5. Negative immunologic assays in a patient with a HEP score ≥ 5 could rule out the diagnosis of HIT. To the best of our knowledge, this is the first study that specifically assessed the diagnostic accuracy of the HEP score in SICU patients. However, our results need to be confirmed in a larger multicenter study, including the HEP score in the algorithm for the diagnosis and treatment of HIT.

## Figures and Tables

**Figure 1 jcm-11-01515-f001:**
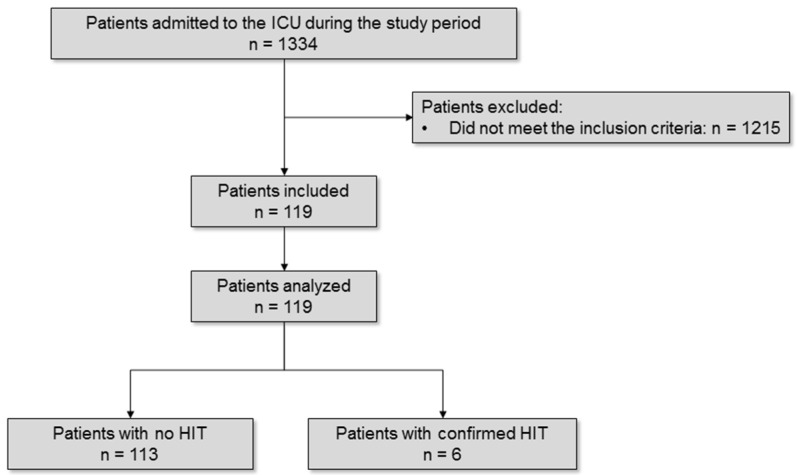
Flow-chart of the study. ICU: intensive care unit; HIT: heparin-induced thrombocytopenia.

**Figure 2 jcm-11-01515-f002:**
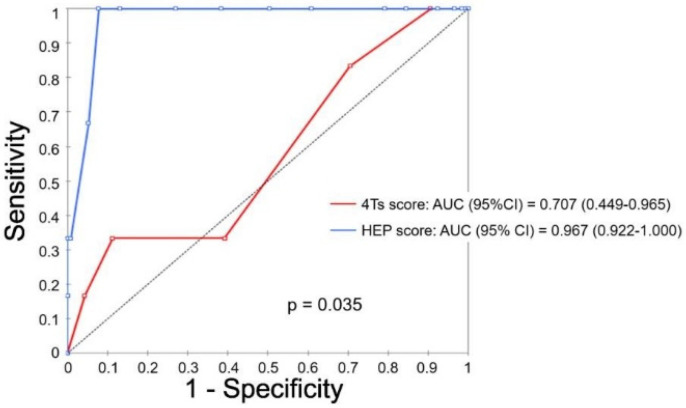
Receiver Operating Characteristic (ROC) curves of the 4Ts and HEP scores. HEP score: Heparin-induced thrombocytopenia Expert Probability score. AUC: area under the ROC curve. 95% CI: 95% confidence interval.

**Table 1 jcm-11-01515-t001:** Baseline characteristics of the study population.

	HIT − (*n* = 113 Patients)	HIT + (*n* = 6 Patients)	*p* Value
Age (years)	66 (59–74)	70 (54–80)	0.62
Male *	80 (71)	6 (100)	0.18
SAPS II	54 (46–68)	46 (38–62)	0.25
Type of heparin *			0.59
Unfractionated heparin	95 (84)	6 (100)	
Low molecular weight heparin	18 (16)	0 (0)	
Reason for heparin therapy *			1.00
Prophylaxis	40 (36)	2 (33)	
Treatment	73 (64)	4 (66)	
Time between heparin therapy and inclusion (days)	4 (2–7)	6 (5–10)	0.16
Platelet count at admission at the ICU (G/L)	164 (121–248)	163 (120–252)	0.50
Platelet count nadir (G/L)	56 (28–98)	56 (26–78)	1.00
Duration of invasive mechanical ventilation (days)	11 (4–20)	17 (8–31)	0.22
Length of stay in the ICU (days)	14 (8–25)	27 (11–42)	0.20
Death in the ICU *	37 (33)	2 (33)	1.00

Data are median (interquartile range). * Data are number of patients (percentage). ICU: intensive care unit; SAPS: simplified acute physiology score.

**Table 2 jcm-11-01515-t002:** Results of the clinical and biological diagnostic tests in the HIT+ and HIT− groups.

	HIT− (*n* = 113 Patients)	HIT+ (*n* = 6 Patients)	*p* Value
**HEP Score**	1 ((−1)−3)	6 (5–9)	0.002
**4Ts Score ***			0.31
0–3	32 (28)	1 (17)	
4–5	68 (60)	3 (50)	
6–8	13 (12)	2 (33)	
**ID-PaGIA ***			<0.0001
Negative	93 (82)	0 (0)	
Indeterminate	4 (4)	0 (0)	
Positive	16 (14)	6 (100)	
**ELISA Test ***			<0.0001
Negative	106 (94)	0 (0)	
Indeterminate	2 (2)	1 (17)	
Positive	5 (4)	5 (83)	
**PAT ***			0.002
Negative	113 (100)	4 (67)	
Positive	0 (0)	2 (33)	
**SRA ***			0.002
Negative	113 (100)	4 (67)	
Positive	0 (0)	2 (33)	

Data are median (interquartile range). * Data are number of patients (percentage). HEP: Heparin-Induced Thrombocytopenia Expert Probability; ELISA: Enzyme-Linked ImmunoSorbent Assay; PAT: heparin-induced Platelets Aggregation Test; SRA: Serotonin Release Assay.

**Table 3 jcm-11-01515-t003:** Diagnostic accuracy of clinical and biological tests in heparin-induced thrombocytopenia.

	Sensitivity	Specificity	PPV	NPV	LR +	LR −
**4Ts Score**						
4Ts Score ≥ 4	0.83 (0.42–0.98)	0.30 (0.23–0.39)	0.06 (0.01–0.11)	0.97 (0.92–1.00)	1.19 (0.82–1.74)	0.55 (0.09–3.38)
4Ts Score ≥ 6	0.33 (0.10–0.70)	0.88 (0.82–0.93)	0.13 (0.00–0.30)	0.96 (0.93–1.00)	2.97 (0.86–10.30)	0.75 (0.42–1.33)
**HEP Score**						
HEP Score ≥ 5	1.00 (0.55–1.00)	0.92 (0.86–0.96)	0.40 (0.15–0.65)	1.00 (1.00–1.00)	12.78 (6.82–23.92)	0
**ID-PaGIA**						
Indeterminate Excluded	1.00 (0.55–1.00)	0.86 (0.78–0.91)	0.27 (0.09–0.46)	1.00 (1.00–1.00)	6.94 (4.41–10.92)	0
‘Indeterminate’ as ‘Positive’ Results	1.00 (0.55–1.00)	0.82 (0.74–0.88)	0.22 (0.07–0.38)	1.00 (1.00–1.00)	5.52 (3.75–8.13)	0
‘Indeterminate’ as ‘Negative’ Results	1.00 (0.55–1.00)	0.86 (0.79–0.91)	0.27 (0.09–0.46)	1.00 (1.00–1.00)	7.25 (4.60–11.43)	0

95% confidence intervals are presented in parentheses. PPV: positive predictive value; NPV: negative predictive value; LR+: positive likelihood ratio; LR−: negative likelihood ratio; HEP: Heparin-induced thrombocytopenia Expert Probability.

## Data Availability

Data are available on simple request to the corresponding author.

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
