# Peer review of "Identification of Heparin-Induced Thrombocytopenia in Surgical Critically Ill Patients by Using the HIT Expert Probability Score: An Observational Pilot Study"

_jcm, 2022, doi:10.3390/jcm11061515_

Round 1

Reviewer 1 Report

the author evaluate the diagnostic accuracy of the HEP score 
for the early identification of HIT in surgical critically ill adult patients. the study is well designed. however, there are some minor points

1- there are different English language error that should be revise with a native English language speaker.

2- in the background, the authors start with the aim of the study, without writing about the background 

3-The order of use of abbreviations is not observed.

Author Response

Dear Reviewer,

Please find enclosed our point-to-point responses to the concerns raised during the round of review of our manuscript. We hope that with these modifications, our manuscript will now be considered suitable for publication in the Journal of Clinical Medicine.

The changes made to the revised version of the manuscript are noted in red.

Thank you for considering the revised manuscript.

Kind regards.

The author evaluate the diagnostic accuracy of the HEP score for the early identification of HIT in surgical critically ill adult patients. The study is well designed. However, there are some minor points.

  1. There are different English language error that should be revise with a native English language speaker.

Response to the Reviewer: The manuscript has been thoroughly revised by a native English-speaking medical writer.

  1. In the background, the authors start with the aim of the study, without writing about the background. 

Response to the Reviewer: the background section of the abstract was modified accordingly, as follows:

“Background: Heparin-induced thrombocytopenia (HIT) remains a challenging diagnosis especially in surgical intensive care unit (SICU) patients. The aim of the study was to evaluate for the first time the diagnostic accuracy of the HIT Expert Probability (HEP) score in the early identification of HIT in SICU patients”

  1. The order of use of abbreviations is not observed.

Response to the Reviewer: Thank you for pointing this out. The list of abbreviations was updated accordingly.

Reviewer 2 Report

This study evaluated the diagnostic accuracy of the HIT Expert 1 Probability score compared with 4T scores in the early identification of heparin-induced thrombocytopenia, which would be useful in guiding clinical decisions regarding therapy.

Suggestions:

  1. In the introduction part, the authors can add background of the challenges of HIT diagnose.
  2. In the introduction part, the authors can briefly summarize how 4Ts and HEP score are calculated, what is main difference between these two methods.
  3. It is difficult to read the content in Table 1&3 due to the space between rows. Please adjust the space between rows.
  4. Font size of Figure 2 needs to be modified.

Author Response

Dear Reviewer,

Please find enclosed our point-to-point responses to the concerns raised during the round of review of our manuscript. We hope that with these modifications, our manuscript will now be considered suitable for publication in the Journal of Clinical Medicine.

The changes made to the revised version of the manuscript are noted in red.

Thank you for considering the revised manuscript.

Kind regards.

This study evaluated the diagnostic accuracy of the HIT Expert 1 Probability score compared with 4T scores in the early identification of heparin-induced thrombocytopenia, which would be useful in guiding clinical decisions regarding therapy.

Suggestions:

  1. In the introduction part, the authors can add background of the challenges of HIT diagnose.

Response to the Reviewer: Thank you for this suggestion. The introduction section has been modified accordingly, as follows (line 21, page 1):

Rapid and early diagnosis of HIT is still difficult. The initial presentation of HIT tend to be limited to the development of thrombocytopenia in a patient under heparin treatment. Laboratory tests prescribed to confirm or rule out HIT suffer from limitations. Immunoassays yield rapid results, with high sensitivity, but with insufficient specificity. Functional assays are unavailable in many hospitals and their sensitivity was recently called into question by expert groups [4,5]”.

  1. In the introduction part, the authors can briefly summarize how 4Ts and HEP score are calculated, what is main difference between these two methods.

Response to the Reviewer: Thank you for this suggestion. The introduction section has been modified accordingly, as follows (line 40, page 2):

The 4Ts and HEP scores are based on items assessing the timing and severity of thrombocytopenia, the onset and timing of thrombosis, the clinical signs of HIT and the potential other causes of thrombocytopenia (see Appendix A, Tables A1 and A2 respectively). The HEP score differs from the 4Ts score in that it weights each item with either positive or negative points (17,18). In particular, the HEP score allows for the attribution of negative points, which may reach -10 when other causes of thrombocytopenia are present, a common situation in SICU patients.

  1. It is difficult to read the content in Table 1&3 due to the space between rows. Please adjust the space between rows.

Response to the Reviewer: The tables were modified accordingly.

  1. Font size of Figure 2 needs to be modified.

Response to the Reviewer: The Figure 2 was revised accordingly.